# Loading and Sustained Release of Benzyl Ammonium Chloride (BAC) in Nano-Clays

**DOI:** 10.3390/ma12223780

**Published:** 2019-11-18

**Authors:** Xianfeng Yue, Rong Zhang, Huairui Li, Minglei Su, Xiaobei Jin, Daochun Qin

**Affiliations:** International Centre for Bamboo and Rattan, Beijing 100102, China; yxf209@163.com (X.Y.); lihuairui2019@jobmail.vip (H.L.); suminglei1122@163.com (M.S.); jinxiaobei@icbr.ac.cn (X.J.)

**Keywords:** Benzyl ammonium chloride, halloysite nanotubes, montmorillonite, sustained release, anti-mildew

## Abstract

Benzyl ammonium chloride (BAC) is a broad-spectrum bactericide, but vulnerable to leaching by water. In this paper, halloysite nanotubes (HNT) and montmorillonite (MMT) were used as drug carriers to load BAC, in order to achieve good anti-mildew activity and long-term sustained release properties. The HNT and MMT nano-composites were characterized by transmission electron microscope (TEM), Fourier transform infrared spectroscopy (FT-IR), X-ray diffraction (XRD), thermogravimetric analysis (TGA), and nitrogen adsorption/desorption. XRD results showed that BAC intercalated the interlayer of MMT, and expanded the interlayer spacing from 1.15 nm to 1.75 nm. Pore analysis showed that BAC decreased the cavity of halloysite nanotubes to a certain extent, which indicated that BAC loaded inside the lumen of HNT successfully. TG analysis showed that the loading capacity of MMT to BAC was higher than HNT. The accelerated-release experiments revealed both two clays have significant sustained-release effects on BAC, and the releasing rate of HNT was relatively lower. Both HNT and MMT have promising application prospects as sustained-release carriers. The inhibition test showed that BAC in nano-clay has good anti-mildew resistance performance.

## 1. Introduction

Quaternary ammonium compounds (QACs) are a kind of cationic surfactants and meanwhile have broad-spectrum bactericide and fungicide properties [1]. QACs with long-chain alkyl groups can attach on to the cell membrane of microorganisms via ionic and hydrophobic interactions. The hydrophobic alkyl chain could invade the lipid bilayer, and destroy the cell wall and membrane [2]. Studies showed that QACs containing 12–18 carbon atoms have good inhibitory effects on bacteria and fungi. QACs are widely used as active ingredients in biomedical materials [3], food hygiene [4], as well as wood preservatives [5]. But the key drawback is that QACs are easily dissolved by water and washed away, although QACs are generally of relatively low toxicity, they can exert significant cytotoxicity to mammalian cells. They also do cause some environmental concern as they can disrupt bacterial ecosystems, especially in the water ecosystem [6]. 

To reduce the release rate and prolong the action time of active substances, sustained release technology has been widely studied in medicine, pesticide, and other fields [7]. The key to achieving sustained release is selecting suitable carriers. Natural inorganic nano-materials, such as clay, have specific characteristics of porous structure and surface activity, and the advantages of environmental friendliness, low cost, and good biocompatibility [8]. Clay and clay minerals are suitable sustained-release carriers due to their high adsorption capacity, high specific surface area, and ion exchange capacity [9]. Clays, such as halloysite (HNT) and montmorillonite (MMT), have been widely studied in drug delivery systems, and proved to have excellent sustained release properties [10].

HNT has a 1-D lumen structure with an inner diameter capable of encasing a broad variety of substances such as drugs, enzymes, and DNA [11,12]. Molecules could be entrapped by HNT in three ways, including adsorption to the external walls, layer intercalation, and loading into the lumen of HNT [13]. Among them, the most essential is the encapsulation of the molecules inside the lumen. HNT also has unique chemical composition with a dual layer of silica and alumina. The alumina on the inner wall could be etched by acid, and the loading capacity could therefore be increased [14]. MMT is a subclass of smectite with similar chemical composition to HNT, and also has a large specific surface area, which exhibits good adsorption ability, cation exchange capacity, stand out adhesive ability, and drug-carrying capability [15]. Unlike HNT, there is no lumen structure in MMT, and MMT loaded drugs between its multilayers. Intercalation of ibuprofen, doxorubicin, and other drugs into montmorillonite has been proved to have good sustained-release properties [16,17]. Both HNT and MMT could be easily modified as nano-materials, which significantly increases their dispersibilities [18].

In this research, two kinds of nano-clays (HNT and MMT) were used as carriers to load benzyl ammonium chloride (BAC), in order to prepare sustained release nano composite with anti-mildew activity. The distribution of BAC in two clays was studied, the release rates of BAC from the two clays in water were investigated, and their anti-mildew efficiency against *Trichoderma viride* and *Aspergillus niger* was evaluated.

## 2. Materials and Methods

### 2.1. Materials

Halloysite nanotubes (HNT) were obtained from Danjiangkou Mineral Factory (Hubei, China), with an average length of 500–1000 nm. Montmorillonite (MMT) was obtained from Usolf Chemical Technology Ltd. (Shandong, China), with a cation exchange capacity of 80 mmol/100 g. Benzyl ammonium chloride (AR) was obtained from Aladdin Industrial Corporation (Shanghai, China). Ethanol (95%) and concentrated sulfuric acid (98%) were obtained from Beijing Chemical Works (Beijing, China).

### 2.2. Preparation of BAC-Loaded HNT

HNT was pretreated via acid etching to increase its lumen volume, and then BAC was loaded by vacuum. The loading mechanism is shown in Figure 1a. The general procedure was: 5 g of HNT were dispersed in 100 mL of 2.0 M sulfuric acid. The suspension was magnetically stirred for 8 h at 80 °C. After reaction, the suspension was precipitated by centrifugation and washed with de-ionized (DI) water until the pH of the supernatant was in the range of 6–7. The final sample was vacuum dried in the oven at 70 °C, the white solid sample was obtained and grinded into powder before use.

2 g of acid etched HNT were dispersed into 100 mL aqua solution of 100 mg/mL BAC. The mixture was sonicated for 30 min to degas, and stirred for 30 min in order to disperse the HNT. Then the suspension was evacuated by vacuum pump. The vacuum state was kept for 30 min and then relieved to atmospheric pressure. This vacuum and pressure-relief process was repeated for three times for a higher loading content. The loaded composites were obtained by centrifugal precipitation and washed by DI water for three times to remove excess BAC. Finally, the solid was dried in the oven at 60 °C until it had a constant weight. The samples were marked as BAC-HNT. 

### 2.3. Preparation of BAC-Loaded MMT

BAC was intercalated into the interlayer of montmorillonite via cation exchange (Figure 1b). The general procedure was: 2 g of MMT were dispersed into 100 mL aqua solution of 100 mg/mL BAC. The mixture was sonicated for 30 min until evenly dispersed. The suspension was then magnetically stirred for 12 h at ambient temperature until it reached adsorption equilibrium. The suspension was obtained by centrifugal precipitation and washed by Deionized water (DI water) for three times to remove excess BAC. Finally, the solid was dried in the oven at 60 °C until it had a constant weight. The samples were marked as BAC-MMT.

### 2.4. Characterization

#### 2.4.1. Transmission Electron Microscope (TEM)

A transmission electron microscope was used to observe the microscopic morphology of HNT and MMT. TEM images were obtained with Tecnai G2 20 (FEI Company, Hilsboro, OR, USA). The sample was ultrasonically dispersed in ethanol for 5 min. A drop of suspension was dropped on to a copper grid and stood for 10 min before being transferred into the microscope.

#### 2.4.2. Fourier Transform Infrared Spectroscopy (FT-IR)

Fourier Transform infrared spectroscopy (FT-IR) was used to characterize the chemical groups of the nano particles. Samples (about 2% weight) were mixed with KBr. Spectra were collected in between 400 and 4000 cm^−1^ with resolution of 4 cm^−1^ and 32 scans by FT-IR Spectrophotometer (Nexus 670, Nicolet Instruments, Waltham, MA, USA).

#### 2.4.3. X-ray Diffraction (XRD)

To determine the structure of modified clay, X-ray diffraction was carried out using X’Pert PRO X-ray Diffraction Analyzer (PANalytical B.V., Almelo, Overijssel, Netherlands). The tube current of the X-ray generator and the working voltage were 15 mA and 30 kV, respectively, with Cu Kα radiation of wavelength at 1.54 Å. The test angle of MMT and HNT was between 2°–10° and 5°–30°, respectively.

#### 2.4.4. Thermogravimetric Analysis (TGA)

Thermogravimetric analysis was used to calculate the loading capacity of BAC in HNT and MMT. TGA was carried out by TA Q500 (TA Instruments, New Castle, DE, USA). Samples weighing between 7–10 mg were heated from room temperature to 800 °C at a heating rate of 10 °C/min under a nitrogen atmosphere with a flow rate of 40 ml/min.

#### 2.4.5. Porosity Analysis

The specific surface areas of HNT and MMT were determined by nitrogen adsorption/desorption using the multiple-point Brunner–Emmet–Teller (BET) method. The Barrett–Joyner–Halenda (BJH) method was used to calculate the total volume of pores and the pore diameter distribution curve. The nitrogen sorption measurements were conducted by automated gas sorption analyzer (Autosorb-IQ, Quantachrome instruments, Boynton, FL, USA), and samples were dried at 120 °C for 12 h under a high vacuum before measurements.

#### 2.4.6. Accelerated Release Test

The release of BAC from loaded HNT and MMT was carried out in vitro by accelerated release. The BAC-loaded HNT and MMT were packed into 3500 Da dialysis bags with a certain amount, respectively. As control, pure BAC with the same amount of loaded BAC in HNT or MMT was precisely weighed and placed into another dialysis bag. Both ends of the dialysis bag were sealed by clips, and then the dialysis bag was immersed into 500 mL deionized water. The liquid was magnetically stirred continuously at room temperature. 1 mL of liquid was collected and sealed in a screw thread vial at regular intervals as a release test sample. 

The BAC content was determined by high performance liquid chromatography-mass spectrometry (HPLC-MS-8040, Shimadzu, Kyoto, Japan), and the release data at each sampling time was collected. HPLC was conducted using a Shimadzu Shim-pack XR-ODS Ⅱ (1.5 mm I.D. × 75 mm L, 2.2 μm, Shimadzu, Kyoto, Japan). The mobile phase was 5 mM ammonium acetate and 0.5% acetic acid solution flowing at 0.5 mL/min. The linear range of BAC concentrations was within 0.001–2.000 mmol/L (R^2^ = 0.999).

#### 2.4.7. Inhibition Test

Two typical fungi, *Trichoderma viride* and *Aspergillus niger*, were used in this experiment to investigate the anti-mildew activity of BAC in nano-clays according to the Kirby–Bauer method [19]. The BAC–clay samples were dispersed respectively in aqueous solution to prepare suspension with 1% wt. concentration of loaded BAC, and the same content of HNT and MMT without BAC was set as the control samples, respectively. Filter paper with a diameter of 5 mm was immersed in the suspension completely, and then air dried at room temperature. The prepared hyphae suspension was evenly coated on the PDA medium. Then the treated filter paper was placed on the medium and each test condition was repeated three times. The fungus was cultured under the condition of 25–28 °C and 85% humidity for one week. The diameter of each inhibition zone was measured by a vernier caliper.

## 3. Results and Discussion

### 3.1. Microscopic Morphology 

The morphology of nano-clay was observed using TEM. Figure 2a,b showed that HNT has a unique hollow tubular structure, with a clear boundary between the wall and lumen, and the relatively smooth internal and external wall. Sulfuric acid could etch the alumina layer in the internal wall of HNT [20], remove some parts of the octahedral alumina layer, and result in a uneven internal surface (Figure 2c). This could also increase the inner diameter of HNT and increase its load capacity [14]. 

Pristine MMT had a multilayer structure (Figure 2d); its layer spacing was less than 1 nm (Figure 2e). After BAC intercalation, the layer spacing of MMT increased significantly (Figure 2f). Due to the ion exchange, the long-chain BAC replaced the metallic cation (Na^+^) and water between the interlayer. Due to a much larger molecular volume of BAC, the interlayer spacing of MMT was expanded.

### 3.2. Thermogravimetric Analysis

Thermogravimetric analysis was used to calculate the loading content of BAC in each clay. The weight loss curve of BAC is shown in Figure 3a. The thermal decomposition of BAC began at 145 °C and it was completely decomposed at 225 °C. There was no residue mass of BAC at 800 °C. 

The thermal decomposition curves of HNT and MMT are shown in the Figure 3b,c. Both HNT and MMT have similar chemical composition, which mainly contain silica and alumina. The mass loss of both clays below 100 °C was mainly caused by dehydration of the absorbed water. The original MMT interlayer contains a much larger amount of adsorbed water, accounting for about 11.20% weight loss of MMT at 100 °C. The weight loss of HNT between 450–650 °C can be attributed to the dehydroxylation of structural Al-OH groups, which is referred as structural water [21,22]. Meanwhile, dehydroxylation of MMT structural Al-OH groups appears between 545–630 °C. After intercalation, most of the absorbed water between the interlayers was replaced by BAC. The new mass loss of BAC-MMT and BAC-HNT in the temperature range of 150–400 °C was due to the decomposition of BAC. It is assuming that the thermal decomposition behavior of loaded BAC is same as the individual. The capacity of BAC in HNT and MMT are calculated as 8.55% and 22.5%, respectively. Limited by the capillary force, BAC was difficult to enter HNT lumen and resulted a relatively less loading content. However, BAC was easily intercalated into MMT via ion exchange. Therefore, MMT has higher loading capacity of BAC [23].

Derivative thermogravimetry (DTG) was applied to study the thermal stability of BAC in nano-clay. Figure 3d delineated the DTG of BAC and nano-clays. It is worth noting that new decomposition peaks (241.7 °C, 314.2 °C and 416.1 °C) appeared in BAC-MMT and replaced the original decomposition peak of BAC at 204.1 °C, which could be attributed to a higher decomposition temperature of BAC between the interlayers of MMT. While in HNT the first decomposition peak of BAC was the same as the original (199.2 °C), and a small shoulder peak appeared at 240 °C, which is similar with the first peak of BAC-MMT. The significant difference of the decomposition behavior of BAC in HNT and MMT was closely related with the loading location, which were the lumen for HNT and the interlayer for MMT. It is speculated that the intercalation of BAC into the interlayer of MMT could limit the thermal motion of BAC molecules, therefore increasing its decomposition temperature. However, the lumen diameter of HNT was too small to accommodate multiple BAC molecules, and some of the loaded BAC aggregated in the orifice; only those inside the lumen would lead to a higher decomposition temperature. 

### 3.3. FT-IR Characterization

FT-IR was used to characterize the chemical structure of halloysite and montmorillonite and the existence of loaded BAC (Figure 4). HNT and MMT have similar chemical structures, both clays are composed of a silicon–oxygen octahedron and an aluminum–oxygen tetrahedron, and the main wavenumbers and assignments of the FT-IR vibration bands were listed in Table 1. As shown in Figure 4, the absorption bands in the 3700–3600 cm^−1^ range were attributed to the inner O–H stretching vibration of aluminum groups. The band at 1031 cm^−^ was attributed to Si–O–Si stretching vibration. The bands in the 700–450 cm^−1^ range were attributed to the stretching vibration of Si–O, bending vibration of Al–O–Si, and bending vibration of Si–O–Si [24]. Compared with HNT, MMT showed a strong absorptions at 3423 and 1639 cm^−1^, which were attributed to the O–H stretching and bending vibration of absorbed water [25]. Corresponding to the TGA results, there was a large amount of absorbed water between the interlayers of MMT; it would be replaced by BAC after intercalation, and lead to weaker absorptions at 3423 and 1639 cm^−1^ in the FT-IR spectrum of BAC-MMT.

After loading with BAC, two new bands appeared in the 2900–2850 cm^−1^ range, which were attributed to symmetric and asymmetric C–H_2_ stretching vibration of the alkyl group in BAC. Another two characteristic bands at 1471 cm^−1^ and 740 cm^−1^ were attributed to C = C stretching vibration and aromatic rings out-of-plane bending vibration [26]. The characteristic peaks of BAC could be seen in BAC–clay samples, which proves the existence of BAC. The peak value of BAC in HNT was relatively weaker than that in MMT, which was due to the lesser loading content of BAC in HNT. The main characteristic peaks of original HNT and MMT were retained. The FT-IR spectra confirmed that the BAC had been loaded into HNT and MMT successfully.

### 3.4. Characterization of Crystal Structure

XRD was used to determine the crystal form before and after BAC loading, whether it intercalated the interlayer of nano clay. Figure 5 shows the XRD patterns of HNT and MMT. The intercalation agent between the layers would cause the increase of interlayer spacing, and lead the X-ray diffraction characteristic peak to shift to a low angle. The location of the first diffraction peak represents the interlayer spacing of nano-clay [27]. The interlayer spacing could be calculated by Bragg’s law [28], as shown below:(1)$$d_{\left(001\right)}=n\lambda /\left(2\sin\theta \right)$$
where *d_(001)_* represent the interplanar spacing, *n* represent the diffraction series and here *n* = 1, *λ* represents the wavelength of the X-ray, and here *λ* = 0.154 nm. 

The XRD pattern of BAC-HNT showed the same reflection peak position with original HNT at 2*θ* angle of 12.11° (Figure 5a), corresponding to a basal spacing of 0.73 nm. This indicates that the long chain of BAC could not intercalate into the interlayer of HNT. Therefore, BAC only filled into the lumen of HNT.

According to Bragg’s law, the basal spacing of MMT and BAC-MMT were 1.15 nm and 1.75 nm, respectively (Figure 5b). The intercalation of BAC in MMT significantly increased the interlayer spacing of MMT. The bond energy between the crystalline layers of MMT was weak and easy to dissociate. Therefore, water or other polar molecules could easily enter the interlayer, break the interlayer bonds, increase the interlayer spacing, and cause the directional expansion of the lattice [29]. In addition, there were some metal cations (Na^+^, Ca^2+^) between the interlayer of MMT, which were easily replaced by the quaternary ammonium cation of BAC, driven by iron exchange. Therefore, according to the XRD result, BAC was successfully inserted between layers of MMT.

### 3.5. Analysis of Specific Surface Area and Aperture Structure

In order to investigate the pore size structure and the BAC loading position of clay, nitrogen adsorption–desorption experiments were carried out. Figure 6 shows adsorption–desorption isotherms of nitrogen for HNT and MMT. The shape of the isotherm gave a qualitative assessment of the porous structure of the materials. The isotherms showed a marked difference in shape. HNT showed an almost reversible Type II nitrogen isotherm (Figure 6b) which indicates that HNT is dominantly macroporous and micropores are absent. MMT showed results like the Type IV isotherms (Figure 6b) and indicated that it contains both mesopores and macropores. The significant hysteresis was caused by mesopores, the plateau in the high pressure was caused by macropores, and the H3 hysteresis pattern indicates the presence of slit-like pores [30].

Pore structure parameters of HNT and MMT are shown in Table 2. Acid etching enlarged the lumen volume of HNT, which increased the average pore size from 15.58 nm to 20.07 nm, so that the loading capacity was increased. As a result, the specific surface area and pore volume increased to 50.22 m^2^/g and 0.290 cm^3^/g, respectively. After BAC was loaded, the specific surface decreased to 28.98 m^2^/g, which was due to the filling and blocking of the HNT lumen by BAC.

After loading with BAC, the average pore size of BAC-MMT increased to 15.56 nm from 6.8 nm. Contrary to BAC-HNT, the loading to BAC in MMT not only blocked the micropores, but also increased the basal spacing via intercalation. The pores in MMT were mainly interlayer micropore and interstitial space between particles. Compare with HNT, the loading content of BAC in MMT were relatively higher. As a result of BAC loading, the specific surface area decreased from 57.51 m^2^/g to 12.24 m^2^/g and the total pore volume decreased from 0.098 cm^3^/g to 0.047 cm^3^/g. These results are in accordance with the above XRD analysis, indicating that HNT could encapsulate BAC inside the lumen, while MMT was intercalated by BAC.

### 3.6. Release Profile of BAC from Nano-Clays

BAC is easily dissolved in water. As shown in the Figure 7, the release rate of BAC in water was very fast, and the release finished in about 5 min. The good solubility may have led to the loss of BAC by water erosion or immersion, which may have shortened its service life for anti-bacterial purposes. 

For both BAC-MMT and BAC-HNT, the release rate of BAC was relatively faster at the initial stage, which may due to the release of adsorbed BAC on the surface or in the orifice. Then the release rate decreased significantly over time, resulting in a sustained release over 24 h. At 24 h, the release amount of BAC-MMT was about 71%. BAC intercalated into the MMT interlayer driven by ion-exchange interaction. The banding force of BAC between the interlayer of MMT was relatively weak. When the water molecules reentered the MMT interlayer, BAC could be dissolved and replaced by water, and released to the outer environment. However, due to the limited transfer channel inside the interlayer of MMT, the release rate of BAC was significantly slowed.

The release of BAC in BAC-HNT was relatively slower. Release was close to equilibrium in 4 hours and the final release amount reached about 30% in 24 h. BAC was loaded inside the HNT cavity. A small amount of BAC may have aggregated in the orifice of HNT, which was easily dissolved by water and caused the initial burst release. But the diameter of the HNT lumens was small and there was a strong capillary force, which hindered the diffusion of the loading agent inside the lumen [31]. Comparing with the 2-dimensional lamellar structure, the HNT only had a 1-dimensional tubular channel for the release of BAC, therefore the release rate was further decreased. 

### 3.7. Inhibition Zone Test

BAC is widely used as component of bacteriostat and anti-mildew agents. Figure 8 showed the inhibition zones of BAC against *A. niger* and *T. viride*, respectively. HNT and MMT were tested as control samples, and no inhibition zones appeared, which illustrated that nano-clay had no anti-mildew activity. BAC-HNT and BAC-MMT formed a clear inhibition zone against both fungus, indicating the BAC had good anti-mildew activity. The inhibition zone diameters of BAC-HNT against *A. niger* and *T. viride* were 11.76 mm and 10.65 mm, respectively. By comparison, the diameters of BAC-MMT were up to 13.22 mm and 12.15 mm. BAC could release from clay and had certain anti-mildew properties. The diameters of the inhibition zone were different, and BAC-MMT had larger inhibition zone diameters than BAC-HNT. Due to the different release rates, BAC was easier to release from the interlayer of MMT. So even if the total amounts of BAC were kept the same between BAC-MMT and BAC-HNT, the effective concentration of BAC-MMT was higher.

## 4. Conclusions

In this work, BAC was loaded into two kinds of clays in different ways, filling into HNT 1-D lumen and intercalating MMT 2-D interlayer. Due to the different loading sites of BAC, the loading capacity, thermal stability, crystal structure, pore structure, and release property had certain differences. Lamellar structure made it easier for drugs to enter and release from MMT. Because of the obstruction of capillary force, although HNT had a large cavity, it was still difficult for BAC to enter and release from the lumen of HNT.

From this work, two kinds of inorganic nano-clay were used as carriers to load BAC, and the effect of sustained release was achieved. This could effectively slow down the release rate of BAC and prolong its action time. Through the inhibition test, it was found that BAC in nano-clay can release and provide good anti-mildew activity. In conclusion, MMT can load more drug, and HNT is more effective in sustained release. 

## Figures and Tables

**Figure 1 materials-12-03780-f001:**
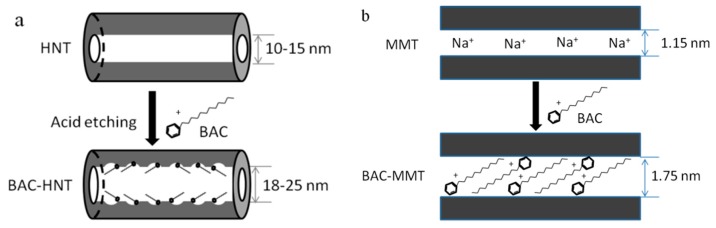
The schematics of (**a**) halloysite nanotubes (HNT) and (**b**) montmorillonite (MMT) loading benzyl ammonium chloride (BAC).

**Figure 2 materials-12-03780-f002:**
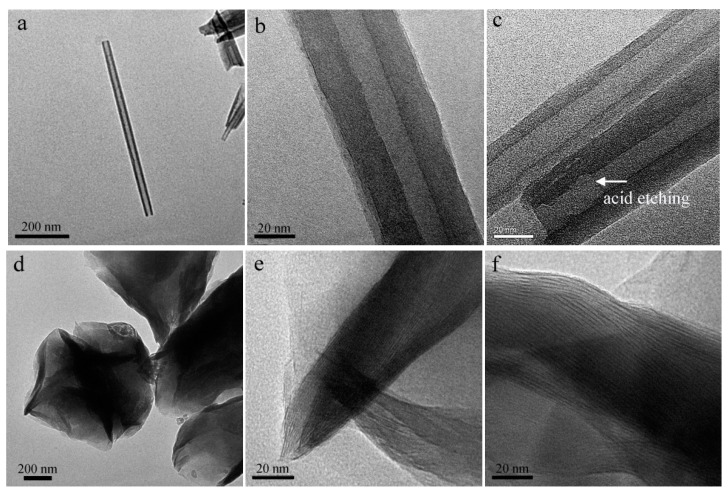
TEM images of (**a,b**) HNT, BAC-HNT (**c**), (**d,e**) MMT, and (**f**) BAC-MMT.

**Figure 3 materials-12-03780-f003:**
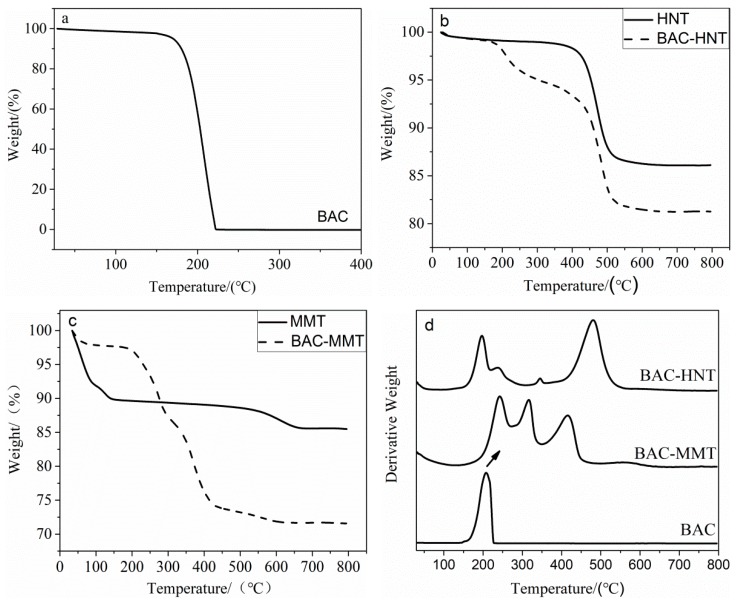
Thermogravimetric analysis: Thermal decomposition curves of (**a**) BAC, (**b**) HNT, and (**c**) MMT; (**d**) derivative thermal curves.

**Figure 4 materials-12-03780-f004:**
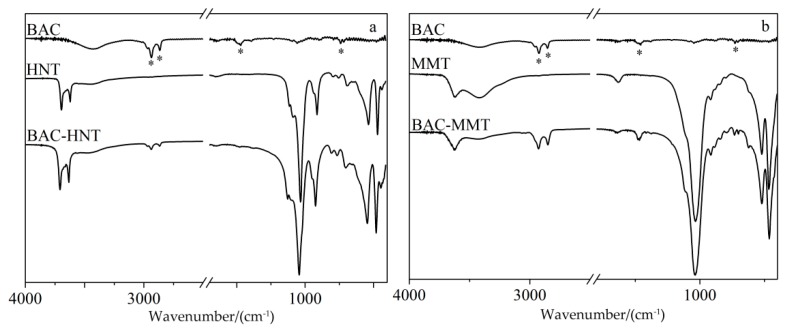
The FT-IR spectra of HNT (**a**) and MMT (**b**) loaded BAC before and after.

**Figure 5 materials-12-03780-f005:**
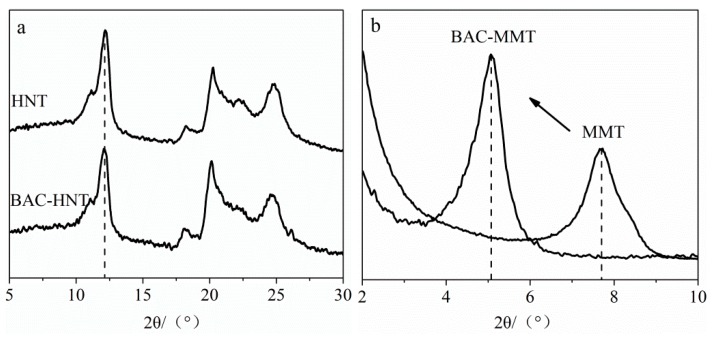
XRD patterns of (**a**) HNT- and (**b**) MMT-loaded BAC before and after.

**Figure 6 materials-12-03780-f006:**
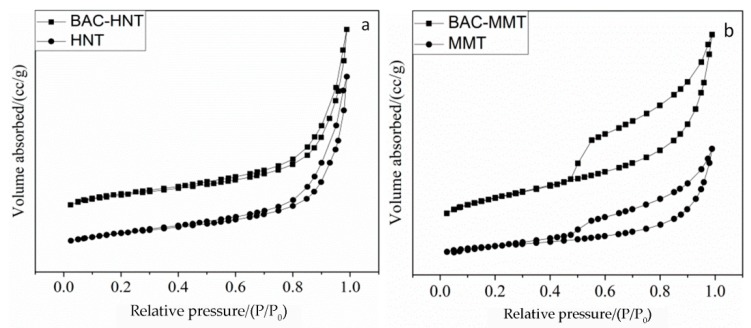
Adsorption–desorption isotherms of (**a**) HNT- and (**b**) MMT-loaded BAC before and after.

**Figure 7 materials-12-03780-f007:**
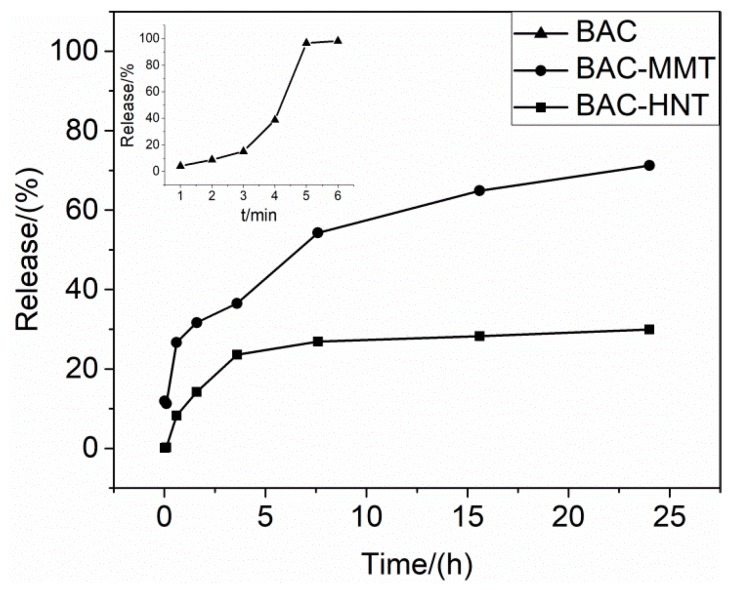
The release profiles of BAC from HNT and MMT.

**Figure 8 materials-12-03780-f008:**
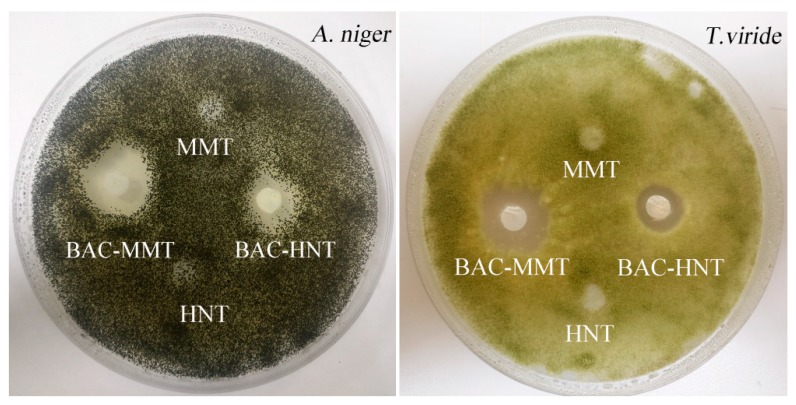
Inhibition zones of BAC-HNT and BAC-MMT against ***A. niger*** and ***T. viride***.

**Table 1 materials-12-03780-t001:** Wavenumbers and assignments of the FT-IR vibration bands.

Benzyl Ammonium Chloride (BAC)		Halloysite (HNT)		Montmorillonite (MMT)	
Wavenumber (cm^−1^)	Assignment	Wavenumber (cm^−1^)	Assignment	Wavenumber (cm^−1^)	Assignment
2900	–CH_2_– asymmetrical stretching vibration	3695 and 3619	inner Al–OH stretching vibration	3625	Al–OH stretching vibration
2850	symmetrical stretching vibration of –CH_2_–	1031	Si–O–Si stretching vibration	3430	H–O–H stretching vibration
1471	C = C stretching vibration	912	O-H bending vibrations	1637	H–O–H bending vibration
740	aromatic rings out-of-plane bending vibration	747 and 797	Al–OH vibrations of the surface hydroxyl groups	1039	Si–O–Si stretching vibration
–	–	534	Al–O–Si bending vibration	522	Si–O–Si bending vibration
–	–	468	Si–O–Si bending vibration	463	Al–O bending vibration

**Table 2 materials-12-03780-t002:** Pore structure parameters of HNT and MMT from N2 adsorption–desorption isotherms.

Sample	Pecific Surface Area/(m^2^·g^−1^)	Total Pore Volume/(cm^3^·g^−1^)	Average Pore Size/(nm)
HNT	43.97	0.138	15.58
Acid-HNT	50.22	0.290	25.03
BAC-HNT	28.98	0.150	20.07
MMT	57.51	0.098	6.80
BAC-MMT	12.24	0.047	15.56

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
