# Peer review of "Loading and Sustained Release of Benzyl Ammonium Chloride (BAC) in Nano-Clays"

_materials, 2019, doi:10.3390/ma12223780_

Round 1

Reviewer 1 Report

This manuscript is relatively well written. The materials and application are of sufficient interest to the journal's readership, and the science as a whole was carried out with appropriate rigor. There are some aspects of the experimental work that should be corrected as discussed below:

Line 30: “QACs have low toxicity to the environment and human beings…” Please provide references for this statement and consider that, although QACs are generally tolerable, they can exert significant cytotoxicity to mammalian cells. They also do cause some environmental concern as they can disrupt bacterial ecosystems in the same way that they kill fungal/bacterial cultures. Figure 2C: What is the white arrow showing? It is difficult for the reader to observe the result of the etching process, and so clearer labeling on the image may help in this regard. Lines 153–154: Are there other reports in the literature that would support the pore expansion theory laid out here? The nitrogen sorption isotherms seem to suggest that the differences may just be due to a change in the surface energy of the material, and not actually the process of enlarging the layer spacings. Figure 4: Please provide peak assignments for the BAC FTIR spectrum. The figure is small, but the two major features appear to be CH2 asymmetric and symmetric stretches at ~2920 and ~2850 cm-1, respectively, which are pretty generic and not really what I would expect the FTIR for BAC to look like. Are there any identifying features for the BAC (e.g., aromatic C-H stretches or ring breathing modes)? Lines 236–240: Microporous materials are defined as materials with pore sizes <2 nm. Therefore, the lack of an inflection in Figure 6A implies that the HNT is a microporous material. Conversely, the MMTs are mesoporous with pore sizes >2 nm. Table 1: The reported pore sizes for the HNTs do not agree with the appearance of the nitrogen sorption isotherm. The sharp rise at >0.8 p/po corresponds to capillary condensation in the voids between HNT particles. Therefore, the calculated pore size in Table 1 is being weighted by this area of the isotherm, and does not actually constitute a pore at all. Preferably, the pore size should be entered as “N/A” for the HNTs since they exhibit a Type II isotherm. For the MMTs, restrict the BJH pore calculation range to p/p0 < 0.8. Also, the corresponding text in the manuscript body should be updated to reflect these changes.

Author Response

Dear Reviewer,

Thank you for your consideration and the reviewers’ insightful comments on the details of our manuscript. The reviewers’ comments and suggestions are very important and helpful for the further improvement of our work. Now we have revised the manuscript (Manuscript ID: materials-624720) entitled “Loading and sustained-release of Benzyl ammonium chloride (BAC) in nano-clays” carefully according to the reviewers’ comments. The detailed corrections and explanations are listed below point by point.

Point 1. Line 30: “QACs have low toxicity to the environment and human beings…” Please provide references for this statement and consider that, although QACs are generally tolerable, they can exert significant cytotoxicity to mammalian cells. They also do cause some environmental concern as they can disrupt bacterial ecosystems in the same way that they kill fungal/bacterial cultures.

Response 1:We thank the reviewer for the suggestion to improve our work. We consulted  relevant literatures on toxicity of QACs, and revised the description in introduction. The high solubility of QACs in water leads to easier loss to the water ecosystem, and might cause series of environmental problems.  In this paper, we mainly focus on reducing the loss of QACs to ensure its safety and environmental concern.

The relative description has been added in revised manuscript.

Saito K , Hayakawa T , Kawabata R , et al. In Vitro Antibacterial and Cytotoxicity Assessments of an Orthodontic Bonding Agent Containing Benzalkonium Chloride. The Angle Orthodontist, 2009, 79(2):331-337. Deutschle T , Porkert U , Reiter R , et al. In vitro genotoxicity and cytotoxicity of benzalkonium chloride. Toxicology In Vitro, 2006, 20(8):1472-1477. 3. Miyauchi, T.; Mori, M.; Imamura, Y. Leaching characteristics of homologues of benzalkonium chloride from wood treated with ammoniacal copper quaternary wood preservative. Journal of wood science 2008, 54, 225-232.

Point 2.Figure 2C: What is the white arrow showing? It is difficult for the reader to observe the result of the etching process, and so clearer labeling on the image may help in this regard.

Response 2: We thank the reviewer point out this shortcoming of our manuscript. Figure 2c has been updated to indicate the arrow meaning. Sulfuric acid treatment makes the inner surface of HNT uneven and it could be attributed to acid etching of alumina layer on the inner surface. As a result, the inner diameter of the lumen increased significantly

Figure 2. TEM images of HNT (a, b), BAC-HNT (c), MMT (d, e), and BAC-MMT (f).

Point 3. Lines 153–154: Are there other reports in the literature that would support the pore expansion theory laid out here? The nitrogen sorption isotherms seem to suggest that the differences may just be due to a change in the surface energy of the material, and not actually the process of enlarging the layer spacings.

Response 3: We thank the reviewer for the suggestion to improve our work. The literature below and our previous research both prove that acid etching method can be used to remove the alumina layer and provide a larger lumen space for loading. According to the literature reported, sulfuric acid treatment provides an efficient method for controllable enlargement of halloysite lumen diameter. Loading efficiency of benzotriazole (corrosion inhibitor) was increased 4 times with such etching technique.

Elshad, A.; Anupam, J.; Wenbo, W.; Yafei, Z.; Yuri, L. Enlargement of halloysite clay nanotube lumen by selective etching of aluminum oxide. Acs Nano 2012, 6, 7216-7226.

Piont 4. Figure 4: Please provide peak assignments for the BAC FTIR spectrum. The figure is small, but the two major features appear to be CH2 asymmetric and symmetric stretches at ~2920 and ~2850 cm-1, respectively, which are pretty generic and not really what I would expect the FTIR for BAC to look like. Are there any identifying features for the BAC (e.g., aromatic C-H stretches or ring breathing modes)?

Response 4: We thank the reviewer for this suggestion. We have identified the characteristic peaks of FTIR and listed the identified bands and their corresponding vibrational modes. By consulting the literature, the characteristic peaks of BAC are described in detail, and the existence of BAC is proved.

The relative description and data have been added in revised manuscript.

Zhou, N.; Qiu, P.; Wang, K.; Fu, H.; Cui, D. Shape-controllable synthesis of hydrophilic NaLuF4:Yb,Er nanocrystals by a surfactant-assistant two-phase system. Nanoscale Research Letters 2013, 8, 518.

Point 5. Lines 236–240: Microporous materials are defined as materials with pore sizes <2 nm. Therefore, the lack of an inflection in Figure 6A implies that the HNT is a microporous material. Conversely, the MMTs are mesoporous with pore sizes >2 nm. Table 1: The reported pore sizes for the HNTs do not agree with the appearance of the nitrogen sorption isotherm. The sharp rise at >0.8 p/po corresponds to capillary condensation in the voids between HNT particles. Therefore, the calculated pore size in Table 1 is being weighted by this area of the isotherm, and does not actually constitute a pore at all. Preferably, the pore size should be entered as “N/A” for the HNTs since they exhibit a Type II isotherm. For the MMTs, restrict the BJH pore calculation range to p/p0 < 0.8. Also, the corresponding text in the manuscript body should be updated to reflect these changes.

Response 5: We thank the reviewer point out this shortcoming of our manuscript. We re-analyzed the pore structure of clay according to our data and the relevant literature. According to the TEM images (figure2), the inner diameter of HNT lumen is about 10-15 nm, which is a macroporous material. And HNT shows an almost reversible Type II nitrogen isotherm (figure 6b) which indicates that HNT is dominantly macroporous and micropores are absent, which is in accordance with the TEM image. MMT do not have the tubular morphology, the layer spacing is only 1-2 nm, so the voids between the MMT particles must be considered in pore structure characterization. MMT shows the Type IV isotherms (figure 6b) and indicates that it contains both mesopores and macropores. We do agree with the reviewer that for MMT the mesoporous and macropores from the experimental data could be attributed to the voids between the MMT particles. The significant hysteresis were caused by mesopores and the absence of the plateau in the high pressure was caused by macropores. And the H3 hysteresis pattern indicates presence of slit-like pores.

The relative description has been added in revised manuscript.

Kuila, U.; Prasad, M. Specific surface area and pore-size distribution in clays and shales. Geophysical Prospecting 2013, 61, 341-362.

Thanks again for the Editors/Reviewers’ warm work and hope that the correction will meet with approval. Should you have any questions, please do not hesitate to contact us.

We look forward to receiving your reply.

Reviewer 2 Report

The manuscript presents an interesting topic. In my opinion, it should be considered for publication after some changes:

The introduction is scarce. It need to be improved and needs to presents the studies that are currently up to date in this field. The results and discussions section needs major improvements. First, any figure needs to be introduced in the manuscript before being inserted. This is not the case in this manuscript. Furthermore, the discussions regarding the results should be also improved and the figures better commented. Moreover, I think that for the FTIR studies a table presenting the identified bands and their corresponding vibrational modes should be added to the manuscript. It would be easier to follow the FTIR section. In addition, some quantitative data of the antimicrobial assays would be useful. Furthermore, the intended application of theses obtained materials require also biocompatibility information. The authors should add to their manuscript some cytotoxic assays of the samples.

Author Response

Dear Reviewer,

Thank you for your consideration and the reviewers’ insightful comments on the details of our manuscript. The reviewers’ comments and suggestions are very important and helpful for the further improvement of our work. Now we have revised the manuscript (Manuscript ID: materials-624720) entitled “Loading and sustained-release of Benzyl ammonium chloride (BAC) in nano-clays” carefully according to the reviewers’ comments. The detailed corrections and explanations are listed below point by point.

Point 1. First, any figure needs to be introduced in the manuscript before being inserted. This is not the case in this manuscript.

Response 1:We thank the reviewer for the suggestion. Now, the relative descriptions have been added in the revised manuscript.

Point 2.Furthermore, the discussions regarding the results should be also improved and the figures better commented. Moreover, I think that for the FTIR studies a table presenting the identified bands and their corresponding vibrational modes should be added to the manuscript. It would be easier to follow the FTIR section. In addition, some quantitative data of the antimicrobial assays would be useful.

Response 2: We thank the reviewer point out this shortcoming of our manuscript. We have identified the characteristic peaks of FTIR and listed the identified bands and their corresponding vibrational modes. The relative description and data have been added in revised manuscript.

Zhou, N.; Qiu, P.; Wang, K.; Fu, H.; Cui, D. Shape-controllable synthesis of hydrophilic NaLuF4:Yb,Er nanocrystals by a surfactant-assistant two-phase system. Nanoscale Research Letters 2013, 8, 518.

Point 3. Furthermore, the intended application of theses obtained materials require also biocompatibility information. The authors should add to their manuscript some cytotoxic assays of the samples.

Response 3: We thank the reviewer point out this shortcoming of our manuscript. We consulted the literature on toxicity of BAC, and revised some descriptions in our manuscript. In this paper, we mainly focus on reducing the loss of QACs to ensure its safety and environmental concern.

Saito K , Hayakawa T , Kawabata R , et al. In Vitro Antibacterial and Cytotoxicity Assessments of an Orthodontic Bonding Agent Containing Benzalkonium Chloride. The Angle Orthodontist, 2009, 79(2):331-337. Deutschle T , Porkert U , Reiter R , et al. In vitro genotoxicity and cytotoxicity of benzalkonium chloride. Toxicology In Vitro, 2006, 20(8):1472-1477. 3. Miyauchi, T.; Mori, M.; Imamura, Y. Leaching characteristics of homologues of benzalkonium chloride from wood treated with ammoniacal copper quaternary wood preservative. Journal of wood science 2008, 54, 225-232.

Thanks again for the Editors/Reviewers’ warm work and hope that the correction will meet with approval. Should you have any questions, please do not hesitate to contact us.

We look forward to receiving your reply.

Reviewer 3 Report

Yue et al. provided very interesting manuscript on Loading and sustained-release of Benzyl ammonium chloride (BAC) in nano-clays. The authors applied 2 materials as the carriers which were loaded with the BAC. The authors applied different methods for the obtained materials characterization such as TEM, X Ray, FTIR, BET TGA . In my opinion the authors should improve Figure 4 and mark the bands (i.e 740 cm-1) which help to follow the spectrum.

I understand that during the Inhibition Zone test, a pure BAC was used as a control? The authors should also provide information on bacteriostat and anti-mildew properties of the HNT and MMT. From the figure 8 it looks like the systems BAC-HNT and BAC-MMT possess the beneficial activities, while the pure BAC does not show them. The activities occur due to the presence of HNT or MMT or due to the BAC? it is not clear from the Figure. The author should re-write this part and higlight what is the control.

Author Response

Dear Reviewer,

Thank you for your consideration and the reviewers’ insightful comments on the details of our manuscript. The reviewers’ comments and suggestions are very important and helpful for the further improvement of our work. Now we have revised the manuscript (Manuscript ID: materials-624720) entitled “Loading and sustained-release of Benzyl ammonium chloride (BAC) in nano-clays” carefully according to the reviewers’ comments. The detailed corrections and explanations are listed below point by point.

Point 1. Yue et al. provided very interesting manuscript on Loading and sustained-release of Benzyl ammonium chloride (BAC) in nano-clays. The authors applied 2 materials as the carriers which were loaded with the BAC. The authors applied different methods for the obtained materials characterization such as TEM, X Ray, FTIR, BET TGA . In my opinion the authors should improve Figure 4 and mark the bands (i.e 740 cm-1) which help to follow the spectrum.

Response 1: We thank the reviewer for the suggestion to improve our work. The relative description and data have been added in revised manuscript. We have identified the characteristic peaks of FTIR and listed the identified bands and their corresponding vibrational modes. By consulting the literature, the characteristic peaks of BAC are described in detail, and the existence of BAC is proved.

Zhou, N.; Qiu, P.; Wang, K.; Fu, H.; Cui, D. Shape-controllable synthesis of hydrophilic NaLuF4:Yb,Er nanocrystals by a surfactant-assistant two-phase system. Nanoscale Research Letters 2013, 8, 518.

Point 2. I understand that during the Inhibition Zone test, a pure BAC was used as a control? The authors should also provide information on bacteriostat and anti-mildew properties of the HNT and MMT. From the figure 8 it looks like the systems BAC-HNT and BAC-MMT possess the beneficial activities, while the pure BAC does not show them. The activities occur due to the presence of HNT or MMT or due to the BAC? it is not clear from the Figure. The author should re-write this part and higlight what is the control.

Response 2: We thank the reviewer to point out the negligence. We apologize for the missing label of HNT in figure 8. We have revised the description of Inhibition zone test to explain the conditions of control samples (Figure 8). Nano clays have no anti-mildew effect on mold. But after loading BAC, nano clays formed obvious inhibition zone, which proves that the existence of BAC provides anti-mold activity for nano clays.

Now, Figure8 is revised as follows and relative description has been added in revised manuscript.

Round 2

Reviewer 1 Report

No further comments; manuscript is suitable for publication at this stage.